# Nanoparticles as Tools to Target Redox Homeostasis in Cancer Cells

**DOI:** 10.3390/antiox9030211

**Published:** 2020-03-04

**Authors:** Francesco Ciccarese, Vittoria Raimondi, Evgeniya Sharova, Micol Silic-Benussi, Vincenzo Ciminale

**Affiliations:** 1Immunology and Molecular Oncology Unit, Veneto Institute of Oncology IOV-IRCCS, 35128 Padova, Italy; francesco.ciccarese@iov.veneto.it (F.C.); vittoria.raimondi@iov.veneto.it (V.R.); evgeniya.sharova@iov.veneto.it (E.S.); micol.silicbenussi@iov.veneto.it (M.S.-B.); 2Department of Surgery, Oncology and Gastroenterology, University of Padova, 35128 Padova, Italy

**Keywords:** nanoparticles, ROS, cancer therapy, redox homeostasis

## Abstract

Reactive oxygen species (ROS) constitute a homeostatic rheostat that modulates signal transduction pathways controlling cell turnover. Most oncogenic pathways activated in cancer cells drive a sustained increase in ROS production, and cancer cells are strongly addicted to the increased activity of scavenging pathways to maintain ROS below levels that produce macromolecular damage and engage cell death pathways. Consistent with this notion, tumor cells are more vulnerable than their normal counterparts to pharmacological treatments that increase ROS production and inhibit ROS scavenging. In the present review, we discuss the recent advances in the development of integrated anticancer therapies based on nanoparticles engineered to kill cancer cells by raising their ROS setpoint. We also examine nanoparticles engineered to exploit the metabolic and redox alterations of cancer cells to promote site-specific drug delivery to cancer cells, thus maximizing anticancer efficacy while minimizing undesired side effects on normal tissues.

## 1. Importance of Reactive Oxygen Species in Cancer Therapy

Selectively killing cancer cells while avoiding systemic toxicity is the primary challenge in cancer therapy. Standard chemotherapeutic drugs and radiotherapy often fail to eradicate tumors and do not discriminate between cancer and highly proliferating healthy cells. Furthermore, most drugs employed in standard chemotherapy are mutagenic and lead to the potential onset of secondary, therapy-induced malignancies in cancer survivors [1,2]. Even therapies designed to target specific signaling pathways altered in cancer cells are not devoid of undesired side effects. The development of novel strategies aimed at increasing the cancer-specific targeting and the therapeutic window of antineoplastic compounds is thus in high demand.

In this context, the elevated levels of reactive oxygen species (ROS) observed in cancer cells compared to their normal counterparts represent a promising therapeutic strategy to target malignant cells selectively [3]. ROS are reactive molecules derived from excitation and univalent reduction of molecular oxygen (O_2_), which lead to the generation of superoxide (O_2_•^−^), hydroxyl radical (•OH) and hydrogen peroxide (H_2_O_2_). ROS are produced in cells by several oxidases and may act as secondary messengers controlling different signal transduction pathways. According to the theory of the ROS rheostat [4], ROS regulate cell fate in a dose-dependent manner (Figure 1). While low/medium levels of ROS promote mitogenic signaling through reversible oxidation of cysteines to sulfenic acid [5] and disulfide bonds [6], high levels of ROS exert cytotoxic effects by inducing base oxidation in nucleic acids and lipid peroxidation, resulting in cell death, which may trigger inflammation and fibrosis. In cancer cells, activation of oncogenic pathways boosts ROS production by the mitochondrial electron transport chain (ETC) [7] and nonmitochondrial oxidases. The increased activity of ROS-scavenging pathways partly curbs such an increase in ROS production. The combined effects of these pathways reset the homeostatic ROS setpoint to a higher level, which provides cancer cells with a proliferative advantage but also makes them more vulnerable to a further increase of ROS that will trigger macromolecular damage and cell death.

Low ROS levels are associated with resting healthy cells (Figure 1, upper left). Physiologic stimulation with mitogenic factors induces an increase in ROS levels, which drive cell proliferation (Figure 1, upper right). Aberrant activation of oncogenic signals results in increased ROS generation, with concomitant upregulation of scavenging systems, which results in a higher ROS setpoint in cancer cells (Figure 1, lower right). NPs are a powerful tool to further increase ROS levels beyond the threshold triggering cell death. Cancer cells are selectively vulnerable to this treatment due to their higher ROS setpoint (Figure 1, lower left).

Consistent with this notion, we recently provided evidence for a ROS-based strategy to selectively kill T-cell acute lymphoblastic leukemia (T-ALL) cells and sensitize them to glucocorticoid-based therapies, while sparing healthy thymocytes [8]. Other evidence points to the anticancer efficacy of therapeutic strategies aimed at inducing oxidative stress [9,10]. Moreover, several anticancer drugs, such as cisplatin [11], doxorubicin [12] and taxanes [13], kill cancer cells partly by increasing ROS levels. Although ROS-inducing compounds are thus likely to be intrinsically selective for cancer cells, their performance could be further enhanced by strategies aimed at confining their damaging activity to the tumor microenvironment.

To this effect, nanotechnologies may provide novel and powerful tools to both alter redox homeostasis in cancer cells and improve the targeting of anticancer drugs to tumor cells by exploiting the unique features of their microenvironment, which include high ROS levels and the acidic pH that results from the glycolytic rewiring of tumor metabolism (Warburg effect). Nanomedicine is based on the use of synthetic particles of 1–1000 nm diameter (nanoparticles, NPs), which can be classified into six main groups: carbon NPs, metal NPs, ceramic NPs, semiconductor NPs, polymeric NPs and lipid-based NPs [14]. In particular, carbon NPs employ the different allotropic forms of carbon, including caged structures (e.g., fullerenes), tubular structures (e.g., carbon nanotubes (CNTs), sheet structures (e.g., graphene and nanodots) [15,16].

## 2. Modulation of Redox Homeostasis by Nanoparticles

Some NPs promote an imbalance in redox homeostasis by both depleting scavenging pathways and by generating ROS [17]. NPs may produce ROS due to their physico-chemical characteristics and the presence of transition metals (e.g., iron or copper) in their composition. Metal-containing NPs promote the generation of ROS via the Haber-Weiss and Fenton-type reactions [18].

In addition to transition metals, the lanthanide cerium also has interesting redox properties, as it can switch between the 4^+^ and 3^+^ oxidation states [19]. Cerium has been employed in nanomedicine mainly in the form of cerium oxide NPs (CeO-NPs), which exhibit redox-modulatory activities that mimic both superoxide dismutase and catalase activities. Interestingly, catalase, but not superoxide dismutase, activity of CeO-NPs is significantly decreased at low pH, suggesting that it could induce oxidative stress and cytotoxicity in the acidic microenvironment that characterizes highly glycolytic tumors, while, at the neutral pH of normal tissues, CeO-NPs could exert antioxidant (i.e., protective) effects [19]. Consistent with this notion, Alili et al. tested the anticancer effects of CeO-NPs in human melanoma cells both in vitro and *in vivo*. Results showed that CeO-NPs induced apoptosis in cancer cells but not in normal tissues and reduced tumor growth and invasiveness in a xenograft animal model in vivo [20]. CeO-NPs were also effective in killing high-grade astrocytoma cells but not normal endothelial cells, which were inhibited in their migratory capacity, suggesting that tumor angiogenesis might also be effectively targeted by this treatment [21]. In addition, CeO-NPs were also used in combination with the anticancer compound doxorubicin, which exerts its cytotoxic effect in part by inducing ROS [12]. Consistent with its dual effect on ROS in normal vs. cancer tissues, CeO-NPs enhanced the anticancer effects of doxorubicin while significantly reducing the undesired side effects of this compound on normal cells, including myocardiocytes. This is a significant finding, given the relevance of myocardiotoxicity of doxorubicin in cancer patients [22]. In a recent study, Hijaz et al. combined CeO-NPs with folic acid to target ovarian cancer cells that overexpress a membrane-bound isoform of folate receptor with cisplatin, which exerts its anticancer activity partly by enhancing ROS levels [11]; results showed a significant anticancer effect both in vitro and in a xenograft mouse model [23].

Small NPs may also enter subcellular organelles and enhance ROS production by activating the ETC and NADPH-dependent oxidases [24]. In this context, super-paramagnetic iron oxide NPs (SPIONs) are endocytosed by cells, internalized in lysosomes and dissociated to release iron ions, which, in turn, translocate into the mitochondria where they target ETC components Fe-S clusters and heme, resulting in O_2_•^−^ production [25]. Similarly, manganese oxide NPs (MnO_2_-NPs) increase ROS levels in isolated mitochondria by affecting ETC complexes I and III, as well as reducing the activity of complexes II and IV, resulting in an impairment of mitochondrial activity [26]. Iron oxide magnetic NPs (IOMNPs) can activate plasma membrane proteins, such as NADPH oxidases [27], inducing the generation of O_2_•^−^ [28]. Arsenic NPs (As-NPs), which are tested for breast cancer treatment [29], increase ROS generation by inhibiting complexes I and II of the ETC [30].

Carbon NPs (e.g., fullerenes, CNTs, carbon nanodots) produce ROS through redox-active functional groups attached to the NPs. In the alternative, transition metal contaminants resulting from the synthesis process of CNTs may catalyze the production of ROS [31]. In addition, due to their large surface area, carbon NPs can absorb other chemicals, which, upon biotransformation, can be oxidized to redox-active quinones and reduced to semiquinones and O_2_•^−^. The latter can be dismutated into H_2_O_2_, which, upon reacting with transition metals, produces hydroxyl radicals [24]. Interestingly, carbon NPs may exert indirect, non-cell-autonomous antitumor effects [32,33], which may result from (i) a decrease of matrix metalloproteinases (MMPs) expression levels, which inhibits tumor metastasis [34] and (ii) enhancement of antitumor immunity [32], partly due to their ability to trigger ROS production by NADPH oxidase and activation of toll-like receptors (TLRs) in professional phagocytes [35]. The use of carbon NPs as photosensors (PSs) in photodynamic therapy (PDT) is described below.

## 3. Nanoparticles in Photodynamic Therapy (PDT)

PDT is a minimally invasive treatment that exploits the physical properties of organic PSs. The majority of the PSs approved for clinical use by the FDA are excited in the far-red spectrum (630–690 nm wavelengths) [36]. PSs are classified into four groups: porphyrins, chlorins, phthalocyanines and a mixed group including hypericin, hypocrellin, indocyanine and methylene blue. The synthetic derivatives of protoporphyrin IX (e.g., taloporfin, temoporfin, verteporfin and photofrin) are the most commonly used PSs in PDT. Unlike radiotherapy, the photons used to trigger the PSs are nonionizing and thus less harmful to the normal tissues [36]. The photoactivation of PSs may initiate two types of reactions termed type I and type II [36,37], both of which generate ROS. Most of the PSs are strongly hydrophobic molecules, which limits their use in the clinic. PSs currently employed in therapy are administrated with a lipid (cremophor) or organic (dextrose, polyethylene glycol) vehicles that are suitable for topical use only. The intravenous inoculation of these pharmacological compositions may lead to severe side effects, including allergic reactions, hypersensitivity and prolonged photosensitization [38]. The suboptimal biodistribution of PSs is another potential problem for routine clinical use. The temoporfin Foscan, a second-generation PS that showed promising results in in vitro experiments, was proposed for the treatment of head and neck cancer. However, Foscan was not approved for clinical use due to its poor tumor selectivity and significant side effects [38]. 

NPs may significantly improve the solubility of PSs, allowing their systemic administration [39]. Cancer-specific targeting of NPs may be obtained, taking advantage of the enhanced glycolytic rate and extracellular acidification that are characteristic of most tumors. Selective targeting of the neoplastic cells can also be achieved by conjugating NPs to metabolites (e.g., folic acid, thiamine), small peptides (e.g., integrin-binding motifs), antibodies, polysaccharides, fatty acids and nucleic acids that either recognize cancer-specific surface molecules or are preferentially taken up by cancer cells [40]. Van Driel and colleagues developed EGF-conjugated NPs containing IRDye700DX as a PS [41]. These NPs were effective in killing head and neck cancer cells in an in vivo mouse model. A similar strategy was used to target leukemia-initiating cells (LIC) with NPs conjugated to anti-CD117 antibodies in a mouse model of chronic myelogenous leukemia [42].

Due to their limited penetration in tissues in humans, the therapeutic use of infrared photons in PDT is limited to skin cancers. However, light delivery in deep tissues was recently obtained using an LED-equipped microdevice implanted in the cancer tissue and switched on using a wireless signal. The emitted light was sufficient to illuminate a volume of 5 mm in diameter and photoactivated a PS in the liver of an adult pig [43].

The beneficial effects of PDT may not be limited to cancer cells only, as ROS produced by photoactivation could activate the immune system and stimulate it to recognize and attack cancer cells. In addition, ROS could damage the blood vessels that feed the tumor, inducing a hypoxic condition [44].

Recently, graphene quantum nanodots (GQNs) were employed as a multifunctional system for both tumor imaging and as electron donors for tumor killing. Tabish et al. showed that, following irradiation with 670 nm photons, GQNs generate singlet oxygen and heat [16]. The resulting combination of photodynamic and photothermal action of GQNs successfully killed MDA-MB-231 breast cancer cells in vitro and in a xenograft mouse model [45]. Efficacy of ROS generation, in part, depends on the presence of functional groups on the surface of nanodots (e.g., ketonic carbonyl, hydroxyl and carboxyl) [46]. In addition, fullerene modified with chitosan and PDT enhanced ROS production by the ETC in human malignant melanoma cells [47].

## 4. Use of Nanoparticles for Redox-Controlled Drug Delivery

Many anticancer drugs have pharmaceutical limitations, including poor solubility, low bioavailability and high cytotoxicity for noncancer cells [48]. A critical application of NPs is to enhance tumor-specific drug delivery while minimizing undesired side effects on healthy cells. To this effect, the high surface area and functional loading capacity of CNTs were successfully employed in vitro and in vivo to deliver anticancer compounds [49] and protect loaded drugs from adverse microenvironment conditions that could inhibit their activity [48].

NPs may be engineered to regulate drug release in response to both exogenous (as in PTD) and endogenous stimuli exploiting the characteristic alterations of tumor metabolism [50]. Doxorubicin linked by disulfide bonds to hyaluronic acid conjugated with single-walled CNTs (SWCNTs) represents an example of a complex redox-sensitive tumor-specific delivery system [51]. Increased levels of hyaluronidase and glutathione in tumor cells [52] trigger hyaluronic acid cleavage and reduction of disulfide bonds, respectively, resulting in the tumor-selective release of doxorubicin [53]. A study from Farjadian and colleagues showed that poly(N-vinylcaprolactam) cross-linked with poly(ethylene glycol) diacrylate forms NPs that release doxorubicin in breast cancer in response to acidic pH and elevated temperature [54].

To overcome the problem of acquired resistance to doxorubicin [55] due to increased extrusion by the ATP-binding cassette (ABC) proteins, doxorubicin was conjugated with bovine serum albumin (BSA) NPs in association with cyclopamine, an inhibitor of the hedgehog pathway, which decreases the expression of ABC proteins. This approach led to the intracellular accumulation of doxorubicin and restored the sensitivity of breast cancer cells to doxorubicin [56]. In a recent work, Yang and colleagues [57] used mesoporous silica NPs (MSNPs), which are internalized by endocytosis, to target a human acute lymphoblastic leukemia (ALL) cell line. MSNPs were loaded with doxorubicin, and their surface was decorated with aptamer Sgc8, which specifically recognizes the protein tyrosine kinase-7 (PTK-7) on the membrane of ALL cells [57]. This system allows specific doxorubicin uptake by leukemia cells, and, as MSNPs release the drug slowly and preferentially under the acidic conditions, it results in sustained drug accumulation at tumor sites, thus enhancing the specific anticancer effect of doxorubicin. Many tumors, including brain, bladder and ovarian cancers, are treated with the platinum-based drug cisplatin, which presents several side effects, such as ototoxicity, nephrotoxicity, hemolysis, neurotoxicity and impaired gametogenesis [58]. Interestingly, platinum NPs (Pt-NPs) accumulate in the tumor site and are internalized into the endosomes of cancer cells, where low endosomal pH and the high ROS levels within cancer cells induce the release of toxic Pt^2+^ ions, leading to DNA damage, cell cycle arrest and apoptosis [59].

Although tumor neo-angiogenesis is a hallmark of cancer [60], tumor vasculature is chaotic and disorganized, leading to inconstant blood flow [61]. Solid tumors are characterized by a high interstitial fluid pressure (IFP), which arises as a consequence of the lack of a functional draining lymphatic system and the elevated density of tumor stroma [62]. The defect of lymphatic drainage in tumor tissues accounts for the enhanced permeability and retention (EPR) effect, which leads to tumor-specific accumulation of macromolecules and NPs [63]. The EPR effect can be enhanced using NPs that release vasodilating agents such as nitric oxide (NO). To this effect, Song and colleagues incorporated polymer functionalized with nitrate into self-assembling d-α-tocopheryl/polyethylene glycol succinate micelles (TNO_3_) that release NO selectively in tumor cells, due to their high levels of glutathione compared to normal cells [52]. Using TNO_3_ loaded with doxorubicin, the authors obtained enhanced delivery of this compound to hepatocellular carcinoma cells in vivo [64], resulting in a synergistic anticancer effect of doxorubicin and TNO_3_ [64]. Furthermore, doxorubicin-derived O_2_•^−^ reacts with NO, generating the reactive nitrogen species (RNS) peroxynitrite (ONOO^−^), a potent oxidant that kills cancer cells [65]. In addition, NO induces the degradation of the anti-apoptotic protein MCL-1, thus activating BAX/BAK-dependent apoptosis [66]. Similarly, Deepagan and colleagues observed boosted EPR effect upon administration of NO-generating NPs (NO-NPs), composed by hydrophilic polyethylene glycol and hydrophobic nitrated dextran, which release NO upon reduction by glutathione [67]. This ensures tumor site-specific vasodilatation that enhances the further uptake of NO-NPs. Doxorubicin-loaded NO-NPs efficiently blocked the growth of HT29 colon cancer cells in vivo by delivering high amounts of doxorubicin specifically to cancer cells. This approach has two interesting implications: (i) glutathione is oxidized during release of NO by nitrated dextran, depowering antioxidant defenses of cancer cells; (ii) although NO is rapidly inactivated to nitrate and nitrite ions, the latter are converted back to NO only in the acidic and hypoxic microenvironment that is characteristic of tumor tissues [68]. Thus, glutathione-sensitive NO-releasing NPs accumulate in tumor tissues due to the EPR effect, promote tumor -specific vasodilatation and drug delivery to cancer cells. The depletion of glutathione, along with the sensitization to ROS-inducing drugs by high levels of NO, maximize anticancer efficacy while sparing normal tissues.

Several other NO-releasing NPs have been developed, including zwitterionic diazeniumdiolate (NONOate)-loaded liposomes, which selectively release NO in the tumor microenvironment [69] due to its low pH. Glucose oxidase (GOx) and l-arginine co-loaded mesoporous organosilica NPs exploit the glucose avidity of cancer cells to generate H_2_O_2_ from GOx-mediated conversion of glucose to gluconic acid and consequent H_2_O_2_-mediated oxidation of l-arginine into NO [70]. These NPs, besides generating a high amount of NO, exert their antitumor effects by inducing glucose starvation of cancer cells [70].

## 5. Conclusions

Rewiring of tumor metabolism and redox homeostasis are critical hallmarks of cancer. In recent years, many studies have built on this knowledge to develop redox-based pharmacological approaches aimed at exploiting the vulnerability of cancer cells to compounds that increase ROS production and inhibit ROS scavenging. The integration of this approach with redox-active NPs (Table 1) may thus prove to be a significant breakthrough in cancer therapy in the near future. To this effect, cerium oxide-based NPs appear especially promising due to their peculiar dual redox properties to induce oxidative stress in cancer cells and exert protective antioxidant effects on normal cells. While photodynamically-triggered NPs appear to be most effective for the treatment of superficial cancers (e.g., skin, head and neck), future developments aimed at delivering photons into deep tissues might significantly extend the use of PTD to many neoplasms. NPs engineered to deliver their cargo in response to acidic pH and glutathione concentration may also prove to be powerful tools to increase the therapeutic window on many anticancer compounds. Rigorous in vivo testing using both genetically-driven and xenograft-based preclinical models as well as clinical trials will be instrumental in defining the real effectiveness and fostering the further development of these combinatorial anticancer strategies.

## Figures and Tables

**Figure 1 antioxidants-09-00211-f001:**
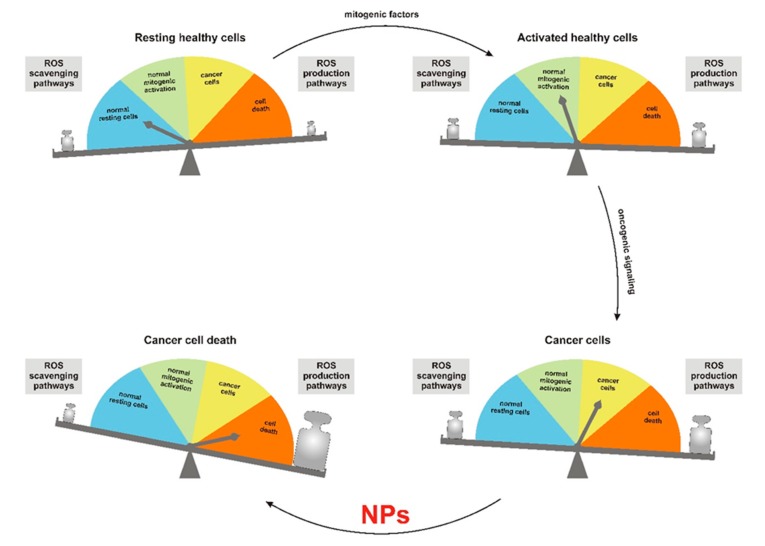
The reactive oxygen species (ROS) rheostat affects cell fate. NPs: nanoparticles.

**Table 1 antioxidants-09-00211-t001:** Summary of the different types of NPs described in the text.

Nanoparticle Systems	Nanoparticle Composition	Mechanism of Action	Model Used for Validation	References
**ROS Modulators**				
CeO-NPs	Cerium oxide	ROS scavenging or generation depending on pH	In vitro cell lines and xenograft animal model	[19,20,21,22,23]
SPIONs	Super-paramagnetic iron oxide	Generation of O_2_•^−^ by the ETC through the release of iron ions	SG-7701, Raw264.7, NIH3T3, HUVEC and HK2 normal cells; N2a, GY7703, HepG2, CNE1 and CNE2 cancer cells	[25]
MnO_2_-NPs	Manganese oxide	Impairment of the ETC and increased ROS production	Evaluation of pharmacokinetics and toxicity of NPs in C57 mice organs	[26]
IOMNPs	Magnetic iron oxide	Activation of NADPH oxidases and generation of O_2_•^−^	Several human cancer cell lines and in vivo animal models	[28]
As-NPs	Arsenic	Inhibition of complex I and II of the ETC	MDA-MB-231, MCF-7 (human breast cancer) cell lines in vitro; isolated rat liver mitochondria	[29,30]
Carbon-NPs	Fullerenes, carbon nanotubes, carbon nanodots	Transition metal-catalyzed generation of ROS and activation of NADPH oxidases and TLRs in professional phagocytes	Human hepatoma (H22) murine model; human pancreatic tumor xenografts mice; U937 (human myeloid lineage cells), nonsensitized human peripheral blood phagocytes	[32,34,35]
**PDT Mediators**				
Ce6	Cerium 6 activated by LED-equipped microdevice	Wireless-activated infrared irradiation	Bladder cancer mouse xenograft and adult pig	[43]
GQNs	Graphene quantum nanodots	Generation of singlet oxygen and heat upon irradiation with 670 nm photons	MDA-MB-231 breast cancer cells, breast cancer xenograft mouse model	[45,46]
**Redox-Based Delivery Systems**				
SWCNTs	Carbon nanotubes + hyaluronic acid + doxorubicin	Release of doxorubicin in the presence of high levels of hyaluronidase and glutathione	MCF-7 (human breast cancer cells), breast cancer xenograft mouse model	[51]
BSA-NPs	Bovine serum albumin + doxorubicin + cyclopamine	Release of doxorubicin and decreased expression of ABC proteins	MDA-MB-231 and MCF-7 (breast cancer cells) in vitro, breast cancer xenograft mouse model	[56]
MSNPs	Mesoporous silica + doxorubicin + Sgc8 aptamer	Specific release of doxorubicin in T-ALL cells after PTK-7 binding	CEM (T-ALL), Ramos (Burkitt lymphoma), Lo2 (normal liver), 293T (human embryonic kidney) cell lines in vitro	[57]
Pt-NPs	Water-soluble platinum NPs capped with polyvinyl alcohol	Release of Pt_2+_ ions at low endosomal pH	Human glioblastoma U251 cell line	[59]
TNO_3_	d-α-tocopheryl/polyethylene glycol succinate micelles + doxorubicin	Release of NO in the presence of high levels of glutathione and synergistic anticancer effect with doxorubicin	Hepatocellular carcinoma cells *in vivo*	[64]
NO-NPs	Hydrophilic polyethylene glycol + hydrophobic nitrated dextran	Boosted EPR effect by releasing NO upon reduction by glutathione	In vitro release of NO in the presence of glutathione, HT29 human colon carcinoma cell line in vitro, HT29 tumor-bearing mice	[67]
NONOate-loaded liposomes	Liposomes + zwitterionic diazeniumdiolate	Release of NO in tumor microenvironment due to low pH	Acellular system with controlled pH	[69]
Mesoporous organosilica NPs	Mesoporous organosilica + glucose oxidase + l-arginine	Release of high amount of NO in the presence of glucose and consequent glucose starvation of cancer cells	U87MG mouse xenograft model	[70]

NPs: nanoparticles; CeO-NPs: cerium oxide NPs; SPIONs: super-paramagnetic iron oxide NPs; MnO_2_-NPs: manganese oxide NPs; IOMNPs: iron oxide magnetic NPs; As-NPs: arsenic NPs; Carbon-NPs: carbon-based NPs; Ce6: cerium oxide NPs under the size of 6 nm; GQNs: graphene quantum nanodots; SWCNTs: single-walled carbon nanotubes; BSA-NPs: bovine serum albumin NPs; MSNPs: mesoporous silica NPs; Pt-NPs: platinum NPs; TNO_3_: d-α-tocopheryl/polyethylene glycol succinate micelles; NO-NPs: nitric oxide-releasing NPs; ETC: electron transport chain; NADPH: nicotinamide adenine dinucleotide phosphate; TLRs: Toll-like receptors; ABC: ATP-binding cassette; T-ALL: T-cell acute lymphoblastic leukemia; PTK-7: protein tyrosine kinase-7; EPR: enhanced permeability and retention effect.

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
