# Peer review of "Nanoparticles as Tools to Target Redox Homeostasis in Cancer Cells"

_antioxidants, 2020, doi:10.3390/antiox9030211_

Round 1

Reviewer 1 Report

In this review, Ciminale et al. consider a variety of recent efforts to develop nanoparticle therapies that target tumor cells using ROS and redox approaches. The manuscript is appropriate in scope, clearly-written, and reasonably well-organized. Along with an introduction, sections are devoted to redox-modulating nanoparticles, nanoparticles used for photodynamic therapy, and nanoparticles that use redox phenomena for tumor targeting. The submission will be of interest to many in the field, but its utility would be significantly improved with some additions.

1) Including a figure demonstrating the “ROS rheostat’ concept and how it differs in tumor vs. normal cells would help the reader to understand the rationale for targeting the related pathways. 

2) A table should be included to help give context to the many nanoparticle systems described by the authors. Within each category (ROS modulators, PDT mediators, and redox-targeted systems), providing information such as nanoparticle composition, carrier system (if applicable), putative mechanism of action, model used for validation (cells or animal model, tumor type), and the relevant references would allow the reader to efficiently compare the types of carriers being reviewed.

Minor point (line 60): The size range for nanoparticles is typically 1-1000 nanometers. Many nanoparticle systems, especially polymer and lipid-based examples, are larger than 100 nm.

Reviewer 2 Report

In this review the authors report recent anticancer therapies based on nanoparticles. The focus is mainly based on the ability of NP to burst the redox state of cancer cells and their use for site-specific drug delivery.

The review is well conceived and clear, however, I suggest to insert some schemes regarding the results depicted in the paragraphs 2 and 4 for a quick highlight of the presented results.

Round 2

Reviewer 1 Report

I appreciate the authors' efforts to improve the manuscript. It is suitable to be published in its current form.